# Analytical Model for Angular Distortion in Multilayer Welding under Constraints

**Woo-Jae Seong** , **Sang-Cheol Park** and **Hee-Keun Lee** *

Industrial Application R&D Institute, Daewoo Shipbuilding & Marine Engineering Co., Ltd., 3370, Geoje-daero, Geoje-si, Gyeongsangnam-do 53302, Korea; wjseong@dsme.co.kr (W.-J.S.); gonausa2005@dsme.co.kr (S.-C.P.)
* Correspondence: zetlee@dsme.co.kr; Tel.: +82-10-3581-8043

**Abstract:** We propose an analytical model for the fast prediction of angular distortion that is caused by practical multilayer (or multi-pass) butt welding under constraints. To this end, the relationships between angular distortion, bead size, thickness, and degree of constraint are derived by analyzing the welding deformation mechanism and considering the bead-on-plate welding experimental results. Prediction curves are then obtained while considering the geometry of the butt welding joint. We verify the formulas through experiments under various constraint conditions, with different welding joint geometries, heat inputs, and thicknesses. The proposed model can not only predict angular distortion in butt joints of various shapes, but also allows for providing restraint methods and welding sequences for minimizing distortion.

**Keywords:** analytical model; angular distortion; multi-layer welding; multi-pass welding; butt welding; degree of constraint; prediction curve; restraint

## 1. Introduction

Welding large structures is essential in constructing ships, heavy machinery, nuclear power plants, platforms, among others. For structures whose mechanical parts require precision work, including rack and pinion jacking systems and large platforms as representative examples, strict criteria should be satisfied during construction. For structures supporting compressive loads, the presence of initial out-of-plane deformation can cause buckling and lead to collapse in the worst case. Deformation in the welding of thick structures is largely due to transverse angular distortion, but is controllable. Therefore, it is necessary to predict and minimize the distortion before construction.

The application of welding technology to industries has been derived in many studies on the relationships between welding conditions and deformation. Okerblom [1] proposed a formula for predicting the angular distortion through a bead-on-plate experiment. The obtained angular distortion was proportional to the heat input and melting efficiency, and inversely proportional to the square of thickness. The heat input is expressed as the product of current and voltage divided by the travel speed. Moreover, Okerblom found that, when penetration in the base metal exceeds 0.6 of its thickness, the base metal under the fusion region fails to provide sufficient constraints, and angular distortion rapidly decreases. Satoh and Terasaki [2] analyzed the relationships between material, heat input, thickness, and deformation. They theoretically determined that the angular distortion is proportional to the heat input and inversely proportional to the square of thickness, and then experimentally verified these results. A common conclusion of the abovementioned studies is that angular distortion first increases in proportion to the division of heat input by the square of thickness, and it gradually decreases after a threshold. Many researchers verified this relationship afterwards, being widely applied across industries. Mochizuki and Okano [3] recently found that this relationship differs, depending on the welding process, and that the factor that actually affects angular distortion is the mechanical melting

region, whose temperature corresponds to that at which the stiffness of the material sharply decreases. Through finite element analysis, they also found that angular distortion is proportional to the width and depth of the region and inversely proportional to the square of thickness, and then confirmed the applicability of these relationships to various types of welding processes.

Besides studies on basic principles, angular distortion has been investigated during multilayer welding. Kihara and Masubuchi [4] determined that angular distortion, which is small at the beginning, increases in the middle section, and then decreases again during multilayer welding of a double-V groove in a ring-type structure. They investigated the effect of the groove shape on the angular change and found the optimal groove height ratio. In addition, they found that back chipping causes a small deformation. Satoh and Terasaki [5] found that angular distortion during multilayer welding depends on the specific deposited heat and derived a formula for predicting deformation from the number of beads based on an existing bead-on-plate welding formulation. They concluded that the number of beads affecting the distortion a function of thickness, heat input, bevel angle, wire density, and specific deposited heat. Kim et al. [6] calculated the angular distortion during the multi-pass welding of stainless steel in the vacuum and cryostat vessel of a fusion reactor by introducing the effective bending rigidity, and proposed a method for minimizing deformation during X groove welding. Ha and Choi [7] derived an analytical formula for estimating angular distortion during V groove multilayer welding while using inherent strain calculation. In addition, they proposed a simple method for predicting the final distortion by considering the measured weld deposition rate. Adamczuk et al. [8] proposed a method for predicting the angular distortion of V groove welding by using curve fitting from the experimental results of butt multi-pass welding. They derived relationships between heat input, geometry, and distortion. Okano et al. [9] conducted a simplified distortion analysis by applying an inherent strain database to improve the efficiency of multi-pass welding analysis and considered the effect of residual stress on existing beads. Seong [10] aimed to apply the characteristics of angular distortion during bead-on-plate welding to multilayer butt welding joints and then proposed a numerical approach while using the geometric principle of weld groove. Consequently, the prediction of angular distortion per pass without using finite element analysis was achieved. Seong et al. [11] introduced the concept of offset and constraint to develop a geometric based algorithm. The angular distortion of 145 mm thick X-groove joint welding was predicted and compared with the finite element analysis results. The optimum welding sequence was provided through the assessment procedure they proposed.

Most of the structures are subject to unknown constraints that are caused by gravity or surrounding structures. Various studies on the degree of constraint and distortion have been conducted to consider these phenomena. Leggatt [12] obtained the constraint strength in terms of thickness, free span of the plate, and experimental constant when both ends are fixed and welded at the middle. A function for estimating angular distortion was derived by dividing the existing angular formula by the restraint factor. Masubuchi and Ich [13] proposed a method for calculating the degree of constraint through computer analysis for various joint configurations. The method might be used for predicting deformation and cold cracking. Ma et al. [14] verified the deformation reduction according to the jig constraint position and pitch through experiments and theoretical analysis, and they addressed longitudinal and transverse shrinkage as well as angular distortion, which were significantly reduced. Kung et al. [15] showed the deformation of SUS304 steel according to the number of jigs using finite element analysis and proposed the optimal location of jig fixtures. Park et al. [16] derived the constraint coefficient through various welding experiments and analyses, obtaining the actual degree of constraint of a ship block and devising the optimal welding sequence. Kim et al. [17] proposed an equivalent strain method while using inherent strain containing functions of the degree of constraint to improve the efficiency of welding deformation analysis for actual ship fabrication, as experimentally verified.

Overall, studies on multilayer welding and the degree of constraint have provided substantial advances. However, few tools have been reported for efficiently predicting angular distortion during multilayer welding while considering the degree of constraint. In fact, it is difficult to reflect unknown

constraints, although they may be designed in advance through laboratory experiments. Finite element analysis has been the mainstream for predicting angular distortion, but its long computational time ranges from a few hours to days, because thermo-elastic-plastic analysis must be conducted for over a few hundred passes. Moreover, the results rely on the experience or intuition of researchers and specialists if analytical methods are not available. In this paper, we propose a method for predicting the angular distortion of a structure under unknown constraints during multilayer welding through a mathematical approach. To this end, a formula on the degree of constraint is derived by analyzing the distortion behavior of bead-on-plate welding. In addition, an analytical formula is derived by applying the first formulation to multilayer welding joint geometry. The calculation of the analytical formula is completed within a few seconds.

## 2. Methods

### 2.1. Mechanics of Welding Distortion

Arc welding uses thermal energy that is converted from electrical energy. Once the molten wire is placed on a base metal, its heat is transferred to the base, and the weld pool cools down, with the base metal undergoing both heating and cooling. The shrinkage of the weld pool is determined by subtracting the tensile plastic strain that is induced by surrounding constraints from the thermal strain during cooling. On the other hand, the shrinkage of the base metal is determined by subtracting the tensile plastic strain that is caused by cooling from the compressive plastic strain induced by surrounding constraints during heating. The constraining effect disappears if the elastic region is removed from the cooled welding joint, and the final contracted strain in the pure plasticity region, known as inherent strain [18], can be obtained. Physically, the inherent strain corresponds to subtracting the elastic strain from the total strain and it equals the sum of the plastic strain and thermal strain if the strain due to phase transformation is negligible, as in low-carbon steel. As the inherent strain at the welding joint includes plasticity, the plastic region is identical to the region where the inherent strain is distributed, and the inherent strain region can be considered as the plastic region.

Angular distortion occurs by the inconsistency between the location of contraction and the neutral axis of the plate. In this study, we aimed to predict angular distortion under constraints due to a connection to surrounding structures. Therefore, a mathematical model for the contraction of the welding joint was devised and experimentally verified. Figure 1 shows the free body diagram of the area where bending occurs due to weld-induced contraction. The figure depicts the force that is caused by contraction in the plastic region, reaction force, and geometrical dimensions. The inherent strain that causes contraction in the bead and base metal is assumed to have a rectangular shape with a uniform distribution. Force $F$ represents the force by contraction, $F_r$ is its reaction force and $M_r$ is the reaction moment by self-equilibrium of the plate without any constraint or external force. When the target structure is connected to other structures and both ends are constrained, $M_c$ is the resulting reaction moment. Thus, the reaction moment that is caused by the equilibrium of force due to the contraction of the welding joint is expressed as the sum of $M_r$ and $M_C$. As we address angular distortion, only the constraints that are caused by the moment in the out-of-plane direction are considered, disregarding the compressive and tensile loads in the in-plane direction. Although the in-plane reaction force at the constraining sides might affect the angular distortion of the welding joint, it is negligible with respect to out-of-plane moment $M_c$. If no constraint by surrounding structures exists, $M_c$ becomes zero, and only the reaction moment by the self-equilibrium, $M_r$, should be considered.

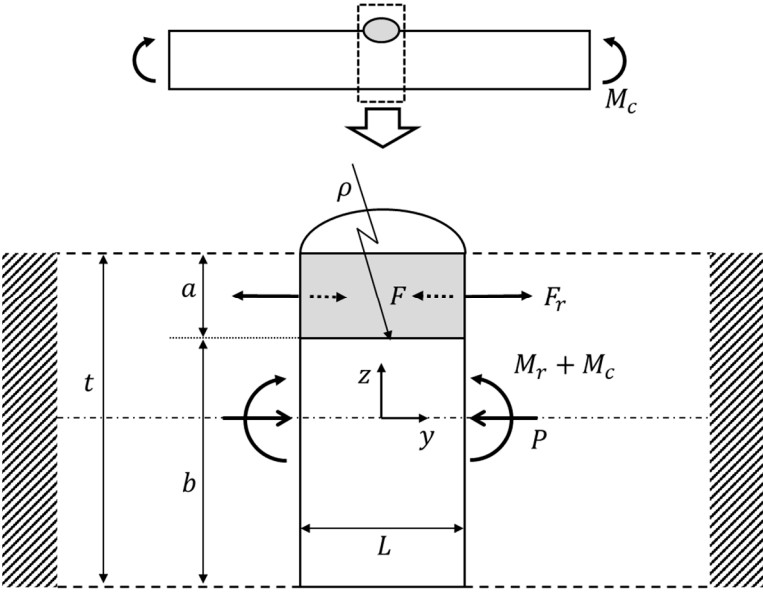

**Figure 1.** Free body diagram of welding distortion (*t*: plate thickness, a: height of plasticity area, *L*: width of plasticity area, $\rho$: curvature, *F*: force by weld shrinkage, $F_r$: reaction force, $M_r$: reaction moment of base plate, $M_c$: reaction moment by surrounding structures, including external moment).

From Figure 1, the equilibrium equations can be expressed, as follows:

$$\sum F = F_r - P = F_r - F = 0 \tag{1}$$

$$\sum M = M_r + M_c - F_r z = 0 \tag{2}$$

The load *F* due to shrinkage and the reaction force $F_r$ against the plastic region beyond the yield strength are in equilibrium. Weighting factor *w* is adopted for the yield strength of the welding joint. The inherent strain that is generated by welding is inhomogeneous in terms of distribution and size due to heat transfer distribution, strain hardening, dilution with welding materials, degree of constraint, and residual stress, but it is higher than the yield strength of the material. Therefore, we introduce the weighting factor to quantify these uncertainties. The ratio of the reaction moment by constraints, $M_c$, to the pure reaction moment by the plate, $M_r$, is defined as $K - 1$. When *K* equals 1, the constraints are removed, because $M_c$ becomes zero and, when *K* is above 1, the degree of constraint increases.

Equation (3) can be obtained by calculating Equations (1) and (2) for the elastic and plastic regions.

$$\sum M = \int_{-\frac{t}{2}}^{\frac{t}{2}} E \varepsilon z dA + (K-1) \int_{-\frac{t}{2}}^{\frac{t}{2}} E \varepsilon z dA + \int_{\frac{t}{2}-a}^{\frac{t}{2}} w Y z dA$$
$$= E \int_{-\frac{t}{2}}^{\frac{t}{2}} \left( \varepsilon_N - \frac{z}{\rho} \right) z dz + \int_{\frac{t}{2}-a}^{\frac{t}{2}} w Y z dz = 0 \tag{3}$$

where $\sigma_z = -\frac{Ez}{\rho}$, $\varepsilon_N - \frac{z}{\rho} = \varepsilon$, *E* : elastic modulus, $\varepsilon$ : elastic strain, $\varepsilon_N$ : elastic strain at neutral axis, $\rho$ : curvature, *w* : weighting factor, *Y* : yield strength, and *K* : degree of constraint, $K = \frac{M_c}{M_r} + 1$. Solving the integral of Equation (3), we obtain

$$\rho = \frac{KEt^3}{6wYab} \tag{4}$$

Geometrically, this can be expressed as the angular distortion that is given by

$$\theta = \frac{L}{\rho - \frac{t}{2}} = \frac{3w\varepsilon_Y b A_p}{Kt^3 - 3w\varepsilon_Y abt} \tag{5}$$

where $\varepsilon_Y$: strain at yield strength, $Y/E$, $A_p$: plasticity area.

We assume that the plate is much thicker than the size of the plastic region. Thus, $3w\varepsilon_Y ab$ is much smaller than the square of thickness $t^2$ and Equation (5) can be expressed as

$$\theta \approx \frac{3w\varepsilon_Y A_p}{Kt^2} \tag{6}$$

where $b \approx t \gg a$, $Kt^2 \gg 3w\varepsilon_Y ab$, which indicates that the angular distortion is proportional to the area of the plastic region and inversely proportional to the square of thickness. The direct relationship between plastic region and angular distortion has been determined in studies on welding and laser forming [2,18,19]. The plasticity of a material is likely to occur at high temperatures between 500 and 800 °C, which produce a sharp stiffness drop. Therefore, the plastic region can be estimated from the distribution of the peak temperature field inside the material during heating and cooling. As we consider a thick plate, the equation of the peak temperature field by the point heat source on the semi-infinite plate is defined, as follows, according to the derivation in [20]:

$$\pi r^2 = \frac{2Q_{\text{net}}}{c_p \lambda e} \frac{1}{T_{\text{max}}} \propto A_{\text{P}} \tag{7}$$

where $r$: distance from point heat source   $Q_{\text{net}}$: effective heat input,   $c_p$: specific heat,   $\lambda$: density, $T_{\text{max}}$: maximum temperature experienced, with $r$ representing the area within an isotherm, because it is the distance from the center of the heat source to a specific temperature. If plasticity occurs at a specific temperature between 500 and 800 °C, the area of plastic region $A_{\text{p}}$ is proportional to the heat input, according to Equation (7).

The wire melting rate is expressed as the sum of the wire resistance heating and the arc heat, which also originate from electrical energy. Thus, for typical arc welding, the bead area is proportional to the heat input. As the plastic region and bead area are determined by the heat input, the angular distortion can be obtained from the bead area, as follows:

$$\theta = \frac{C}{K} \frac{A}{t^2} \tag{8}$$

where $A$: bead area, $C = 3w\varepsilon_Y \cdot A_p / A$.

Equation (8) indicates that the angular distortion by welding is proportional to the bead area and inversely proportional to the square of thickness and degree of constraint only for materials with sufficient thickness when compared to the heated part. Here, the degree of constraint $K$ must be 1 or above, and constant $C$ depends on the material properties of the base metal and wire, the wire diameter, and the welding process. This constant includes errors while deriving Equations (1) through (8), and its value can be experimentally obtained through bead-on-plate welding for $K = 1$ without the external constraints. In this study, we maintain the same material and welding process to keep the constant $C$ unchanged.

## 2.2. Bead-on-Plate Welding Experiment

We conducted an experiment of bead-on-plate welding to verify the validity of Equations (1) to (8) while considering various assumptions. Low-carbon steel AH32 was used as the base metal, and E81T1-K2C flux cored wire with 1.2 mm in diameter and meeting the American Welding Society (AWS) Standards was used as welding material. The shielding gas was 100% $CO_2$. Welding was performed at two trisection lines in a 1500 × 1000 mm specimen with different thicknesses without constraints, as shown in Figure 2. The thicknesses of the bead-on-plate welding specimen were determined in consideration of the bead-formed thickness generated during the multilayer welding. The width was larger than the length along the welding direction to minimize the influence of longitudinal distortion. The specimen size and measurement method were adopted, as in the previous study [10].

Constant current and voltage were maintained, but the welding speed was varied to achieve different heat input conditions. Table 1 lists the experimental conditions.

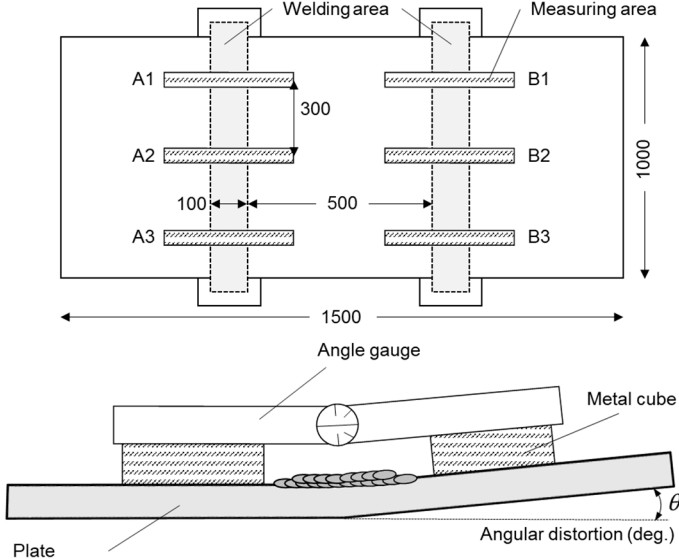

**Figure 2.** Bead-on-plate welding and measurement (unit: millimeters) [10].

**Table 1.** Welding conditions for bead-on-plate welding experiment.

| Thickness (mm) | Voltage (V) | Current (A) | Travel Speed (cm/min) |
| --- | --- | --- | --- |
| 11.5, 12, 19.5, 45 | 29 | 285 | 25, 30, 40, 60 |
| 28 | 29 | 285 | 25, 30 |

We performed multi-pass welding up to two layers to increase the measurement accuracy of the bead cross-sectional area and height. Welding was performed twice per condition, and the angular distortion was measured for each pass of the first layer. For angular distortion, the average values were obtained at three points (A1, A2, A3 or B1, B2, B3 in Figure 2) while using a digital angle gauge. Metal cubes were used to facilitate measurements.

Figure 3 shows the bead cross-section after welding. The bead cross-sectional area stacked on the base metal was obtained by calculating the number of pixels. The maximum and minimum bead heights were obtained by creating curves parallel to the deformed geometry of the base metal, with their average being considered to be the bead height.

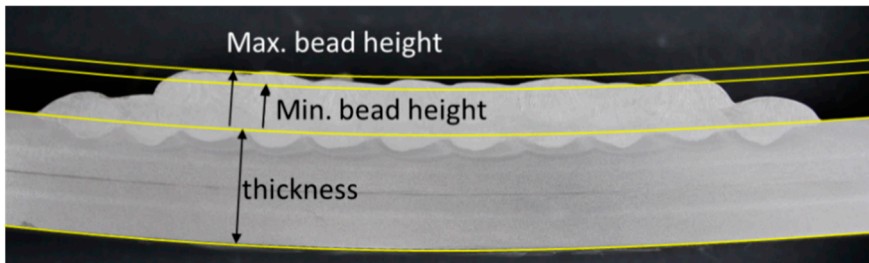

**Figure 3.** Magnified image of welded sample for welding at 285 A, 29 V, and 30 cm/min

Figure 4 shows the measured angular distortion for varying thicknesses of the base metal. In a base metal with constant thickness, the angular distortion linearly increases with the number of passes. As the base metal thickness increases, the angular distortion reduces and, hence, the angular distortion per pass is constant at a given thickness.

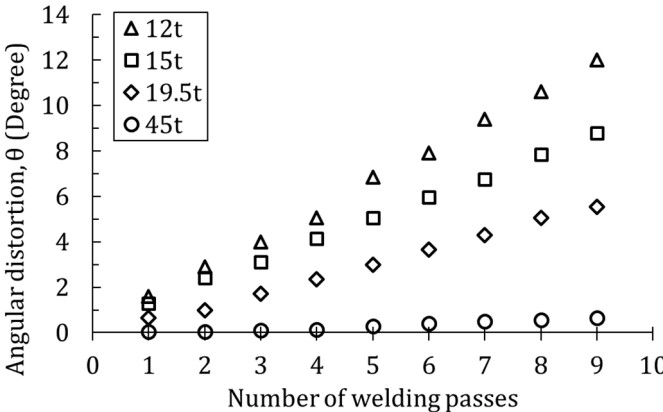

**Figure 4.** Angular distortion according to number of welding passes.

Figure 5 shows the measured bead cross-sectional area and square of bead height per pass, according to the heat input. As the heat input increases, the bead area and height proportionally increase. We assumed that the bead heights that were obtained during multilayer welding were the same as those measured in this experiment under the same conditions.

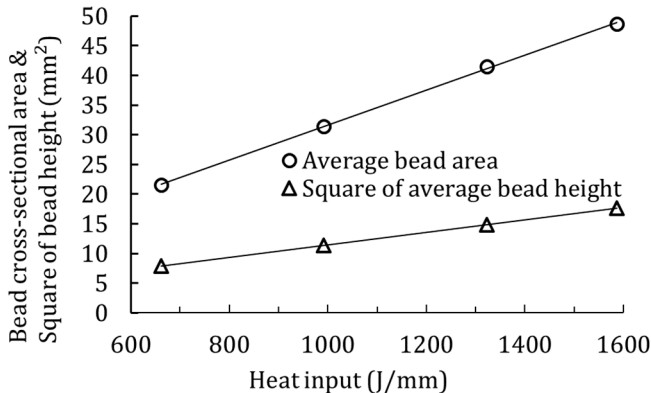

**Figure 5.** Bead cross-sectional area and square of height according to heat input.

Figure 6 shows the relationship between the ratio of bead area to the square of base metal thickness and angular distortion. The angular distortion increases proportionally to the ratio, but decreases after a threshold, which was found to be approximately 0.375. The threshold seems to occur as the center position of the contracted load becomes closer to the neutral axis. The data above the threshold produce low measurement accuracy of angular distortion due to in-plane shrinkage, and they exhibit nonlinearity [2,3]. If the center position of the shrinkage load agrees with the neutral axis, the angular distortion becomes zero, and shrinkage reaches its maximum. Reverse angular distortion might occur if it is located below the neutral axis. The reason is that if the thickness under the beads is not sufficient, the temperature gradient can be reversed, depending on the ambient cooling conditions. In addition, the reverse distorted shape generated during heating might be maintained until being completely cooled down. Xie et al. [21] has reported that the distortion direction can be affected by the asymmetry of the cross-sectional profile along the thickness direction and even pores generated by insufficient flow of the molten pool. Consequently, it is almost impossible to quantitatively predict the amount and direction of deformation above the threshold. Therefore, we only address cases in which the distance between the load position and neutral axis is sufficiently long, as assumed in Equation (6). Thus, we used data that were below the threshold (dashed line in Figure 6). When all of the the welding passes are completed at the same heat input, the third layer or beyond results in welding within this range.

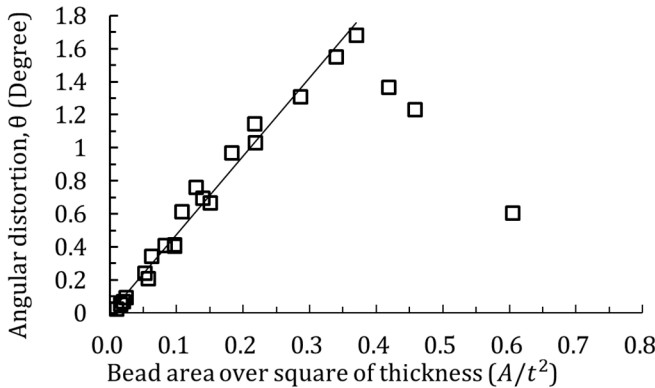

**Figure 6.** Relationship between angular distortion and ratio of bead area to square of thickness.

Without external constraint, the calculated value of $C$ in the linear section below the threshold was 4.75 from Figure 6. Hence, we verified the relationship between bead area and angular distortion in Equation (8). The relationship between heat input, bead area, and angular distortion, as shown in Figures 5 and 6, implies that heat input is proportional to plastic area as in Equation (7), which is proportional to angular distortion shown in Equation (6).

### 2.3. Multilayer Welding Considering Degree of Constraint

We derived the relationship between the bead area and angular distortion through a mechanical approach and experiments, as shown in the previous sections. We also aimed to obtain the analytic solutions for practical butt welding joints. As multilayer welding is a repetitive process, the welding conditions do not notably change across layers and, hence, we assumed a uniform amount of weld deposition. As the accumulation of beads increases the thickness, it complicates distortion across layers. The bead area and thickness for a heat input can be calculated while using the relation that is shown in Figure 5. Thus, the angular distortion can be predicted if the geometry of the welding joint is known. The total angular distortion at a layer can be obtained from the total bead area, as the angular distortion linearly increases with the number of passes at a given thickness (Figure 4). Equation (9) is obtained by applying Equation (8) to a V groove welding joint, as shown in Figure 7.

$$\theta_{\text{total}} = \theta_0 + \sum_{i=1}^{n+1} \Delta\theta_i = \theta_0 + \sum_{i=1}^{n+1} \frac{C}{Kt^2}\Delta A_i = \theta_0 + \sum_{i=1}^{n+1} \frac{C}{Kt^2}|y_{\text{R}}|\Delta z + \sum_{i=1}^{n+1} \frac{C}{Kt^2}|y_{\text{L}}|\Delta z \tag{9}$$

where $\theta_{\text{total}}$: total angular distortion, $\theta_0$: initial angular distortion, $\Delta\theta_i$: angular distortion by layer $i$, $i$: layer number after initial thickness, $n$: total number of layers, $t$: thickness, $\Delta A_i$: area of layer $i$, $y_{\text{R}}$, $y_{\text{L}}$: linear equations of right and left bevel groove, respectively, $\Delta z$: thickness of layer, heat height.

In Equation (9) with Figure 7, $\theta_0$ and $h_0$ are the initial angular distortion and initial thickness, respectively. The initial angular distortion is the angle at the initial thickness, and the values should be determined after welding at least two layers. In typical weld groove joints, rather than welding onto the base material, the accumulation of beads (layers) creates the thickness. The angular distortion at the first and second layers cannot be accurately predicted, because the layers belong to conditions above the threshold, as shown in Figure 6. In addition, some distortions occur during initial setting and fit-up. Hence, the initial thickness and its angular distortion should be considered to include these errors, but the purely predicted angular distortion excludes the initial angular distortion.

Layer number $i$ is zero at the initial angular distortion, and $n$ denotes the total number of layers stacked up to the thickness of the base material. This thickness is obtained by subtracting the initial one from that of the base material and dividing it by the bead height. $n + 1$ was set by adding a layer, as the surface of the final bead layer must be higher than the base metal thickness. The width of a layer,

$|y_L| + |y_R|$, represents the length from the left to right bevel surfaces, as shown in Figure 7. The width can be geometrically obtained from the linear equation of each bevel surface in the $y$–$z$ plane. The bead area of layer $i$ is then obtained by multiplying the width by layer height $\Delta z$.

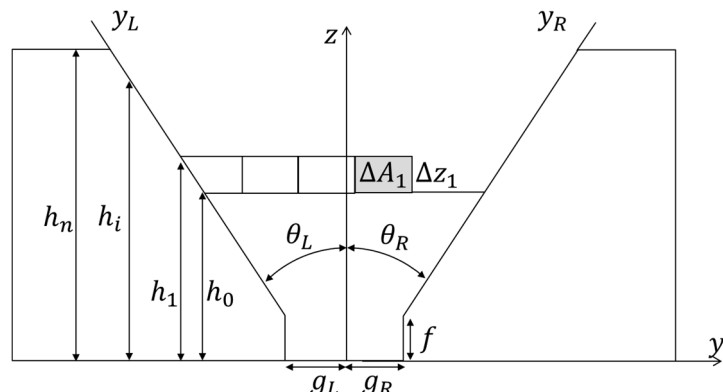

**Figure 7.** Geometry of V groove welding joint ($f$: root face, $g_L$, $g_R$: left and right root gaps, respectively, $h_n$: thickness of base metal, $h_0$: initial thickness, $\theta_L$, $\theta_R$: left and right bevel angles, respectively).

The angular distortion after the initial one can be calculated, as follows:

$$
\begin{aligned}
\sum_{i=1}^{n+1} \frac{C}{Kt^2}\big(|y_L + y_R|\big)\Delta z \\
= \frac{C}{K} \sum_{i=1}^{n+1} \frac{C}{Kz^2}[\tan\theta_R \cdot (z - f) + g_L + \tan\theta_L \cdot (z - f) + g_R]\Delta z
\end{aligned}
\tag{10}
$$

where $h_0$: initial thickness, $y_R = \tan\theta_R \cdot (z - f) + g_R$, $y_L = \tan\theta_L \cdot (z - f) + g_L$, $t = z$.

Substituting Equation (10) into Equation (9), the total angular distortion of a V groove can be obtained, as follows:

$$
\theta_{\text{total}} = \theta_0 + \frac{C}{K} \sum_{i=1}^{n+1} \frac{[\tan(\theta_R)_i + \tan(\theta_L)_i] \cdot [(h_0 + (i-1)\Delta z - f) + g_L + g_R]}{(h_0 + (i-1)\Delta z)^2} \cdot \Delta z
\tag{11}
$$

where $z = h_0 + (i - 1)\Delta z$.

Equation (11) represents the sum of the initial measured angular distortion and the predicted angular distortions at all subsequent layers. Angular distortion decreases as constant $C$, root gap, and bevel angle decrease, and the degree of constraint $K$, initial thickness, and root face increase. The angular distortion during V groove welding depends on geometric parameters, except for two factors, namely, bead height $\Delta z$, which is determined by the heat input (Figure 5), and constant $C$, which is experimentally determined (Figure 6).

Figure 8 shows the angular distortion prediction curves that were obtained from Equation (11). The position of the starting point of the curve and its slope must be determined for a prediction curve to be determined. The starting point is obtained from initial thickness $h_0$ and initial angular distortion $\theta_0$, whereas the slope is obtained from degree of constraint $K$, which requires another point after the starting point. In Figure 8, the initial thicknesses of curves A and B are $x2$ and $x1$, and their initial angular distortions are $y1$ and $y2$, respectively. Although curves A and B have different initial angular distortions and thicknesses, they are almost parallel with the initial and final differences in angular distortion being the same. In practice, this results from the distortion by fit-up, presetting, and welding up to the second layer. Curves B and C have the same starting point, but different slopes. The slope of the curve becomes steeper as the degree of constrain reduces and the root gap or bevel angle increases with decreasing root face. Under the same geometry, curve C has a lower degree of constraint $K$ than curve A or B.

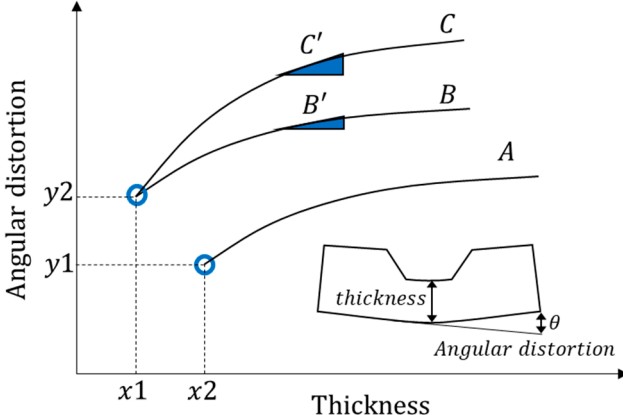

**Figure 8.** Prediction curve of angular distortion based on Equation (11).

In addition, we considered that the angular distortion of each layer reduces the bevel angle. Hence, the amount of deposition decreases at the subsequent layer, which further decreases the angular distortion. The predicted angular distortion and bevel angle at the previous layer must be subtracted to obtain the bevel angle at a layer, as shown in Equation (12). This is used as the bevel angle term in Equation (11).

$$(\theta_R)_i = (\theta_R)_{i-1} - (\Delta\theta_R)_{i-1}, \ (\theta_L)_i = (\theta_L)_{i-1} - (\Delta\theta_L)_{i-1} \tag{12}$$

where $(\theta_R)_i$: right bevel angle at layer $i$, $(\Delta\theta_R)_{i-1}$: angular distortion at layer $i-1$, $(\theta_L)_i$: left bevel angle at layer $i$, $(\Delta\theta_L)_{i-1}$: angular distortion at layer $i-1$.

Equation (11) for predicting the angular distortion has three unknown variables, namely, $\theta_0$, $h_0$, and $K$. Initial thickness $h_0$ and corresponding angular distortion $\theta_0$ can be determined from measurements at the desired layer, but at least two layers should be stacked. The degree of constraint $K$ can be obtained by measuring the angular distortion of the additionally accumulated layer after the first measurement (the starting point in Figure 8). For calculation, we increase $K$ by 0.01 from 1.0 and adopt the value that agrees with the measurement after two or more layers have been accumulated after the first measurement. Alternatively, it can be directly obtained when the angular distortion is known after only one layer is stacked at the first measurement, by using $i = 1$ in Equation (11) and the following expression for $K$:

$$K = \frac{C \ (\tan\theta_R + \tan\theta_L)\cdot[(h_0 - f) + g_R + g_L]\Delta z}{h_0^2(\theta_1 - \theta_0)} \tag{13}$$

Once $K$ is determined, it can be used to predict the angular distortion at all subsequent layers. However, $K$ must be recalculated following the abovementioned procedure if the constraint condition changes during welding.

## 3. Results and Discussion

### 3.1. Validation of Estimation

We conducted V butt welding experiments to examine whether the predictions that are based on Equation (11) agree with experimental values. The size of the specimen was $500 \times 1000$ mm and its thickness was 28 mm, as shown in Figure 9. Temporary hexahedron members were fit-up by tack welding at points where welding started and ended. The joint groove had a 4 mm root gap and a 50° groove angle. Multi-pass welding was then performed, and the thickness and angular distortion were measured at each layer. One strongback was installed in the center of the specimen, and two at each end where transient heat transfer and deformation occurs. The experiments were performed

with and without strongback, while the other conditions were the same as those for the bead-on-plate welding experiment.

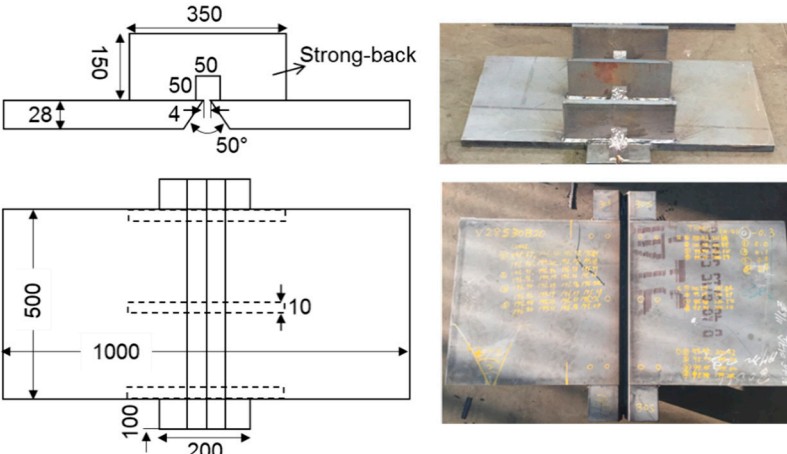

**Figure 9.** Specimen for V butt welding experiments (unit: millimeters).

In this experiment, the thickness that was generated up to the second layer was adopted as initial value $h_0$, and the corresponding angular distortion was set as initial value $\theta_0$. They were determined by measurement. The degree of constraint $K$ is calculated from the angular distortion of the second measurement, but, in this section, we derived the $K$ value that best matches all of the measured data for the verification of the equation. Table 2 lists the parameters for prediction by Equation (11). Figures 10 and 11 show the prediction curves and experiment results. In Figure 10, the value of $K$ in the free (unconstrained) condition was calculated to be 1.09 (close to 1.0). We minimized the contribution to the restraint increase, as temporary pieces were tack-welded at both ends and the grooves were cut in the same way as specimens. When constrained by a 10 mm strongback, $K$ was calcucated to 10.5. Figure 11 shows the results for different strongback thicknesses. The degree of constraint $K$ was 13.5 for a 20 mm strongback. Although the thickness of the strongback doubled, the degree of constraint increased by 1.3 times. When compared to the case with no strongback, the change in angular distortion decreased to 1/11 for 10 mm strongback and 1/13 for 20 mm. Hence, we found that the fixture itself had a greater effect on the increase in the degree of constraint than the fixture thickness.

**Table 2.** Input parameters for angular distortion estimation.

| Parameter | No Strongback | Strongback 10$t$ | Strongback 20$t$ |
|---|---|---|---|
| $\theta_0$ | 0.05 | 0.3 | 0.1 |
| $h_0$ | 8.7 | 10 | 9 |
| K | 1.09 | 10.5 | 13.5 |

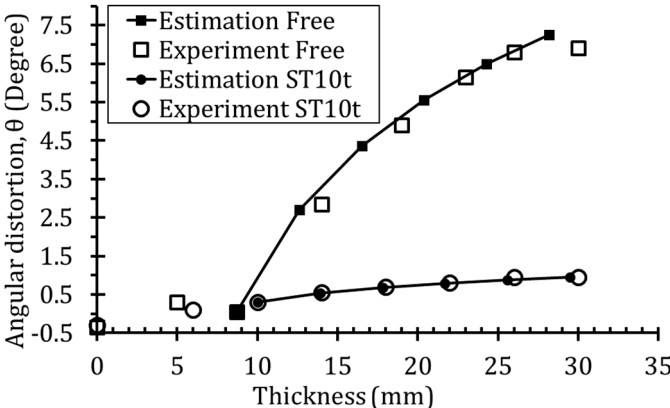

**Figure 10.** Comparison of experimental and predicted values in free (unconstrained) and constrained states for welding conditions of 285 A, 29 V, and 30 cm/min (ST10t: strongback with 10 mm thickness).

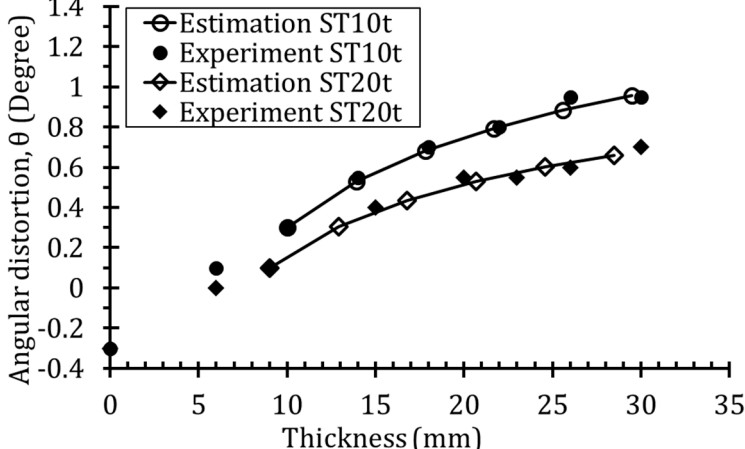

**Figure 11.** Comparison of experimental and predicted values under constraints for welding conditions of 285 A, 29 V, and 30 cm/min (ST*xx*t: strongback with *xx* mm thickness).

These results indicate that once the appropriate degree of constraint is determined from the values measured at a layer after initial thickness, the subsequent angular distortion can be accurately predicted. Thus, we verified the applicability of Equation (11) regarding the constraint effects.

### 3.2. Applications

We applied the proposed prediction method to a plate with 50 mm in thickness. The current and voltage were set to 280 A and 30 V, respectively, and the groove angle and root gap were set to 35° and 5.7 mm, respectively. The thickness and angle were measured at each layer. The other experimental conditions were the same as those for the V groove experiment (Section 3.1). For travel speed of 25 cm/min, the initial thickness and angular distortion were 13 mm and 1.9°, respectively, upon the completion of the second layer. The measured angular distortion after welding of the third layer was 3.35°. For a welding speed of 60 cm/min, the initial thickness and angular distortion were 13 mm and 0.15° after third layer with four passes, respectively, and the measured distortion at the next layer was 1.23°. In this case, the change of the angular distortion was so small at the beginning that the measurement was started from the third layer. In these two cases, $K$ was directly calculated while using Equation (13), because the results before and after welding one layer were known, with the respective values being 1.14 and 1.47.

Figure 12 shows the prediction curves, including subsequent layers. No strongback constraint was considered in both specimens, but the calculated degree of constraint was above 1.0. Hence,

some constraints were applied to the welding joint. Specifically, the constraining effect by self-weight seems to have acted on the welding joint. The 50 mm thick test specimen was 1.8 times heavier than the 28 mm thick test plate above. Moreover, the thick temporary pieces that were attached to the starting and ending points of welding may have partially contributed to the increase in degree of constraint. A small heat input due to fast welding exhibited a higher degree of constraint than a large heat input. The constraining effect by self-weight seems to have been larger when the amount of weld deposition was smaller due to the smaller heat input. Murakawa [18] defined the degree of constraint as the ratio of stiffness of the surrounding structure to sum the stiffness of the welding joint and the surrounding structure through the bar-spring model, which is commonly used as a welding mechanism. In summary, when the bead is small due to small heat input, the degree of constraint increases by the relatively small stiffness of the welding joint when compared to the stiffness of the surrounding structure, as confirmed in this study.

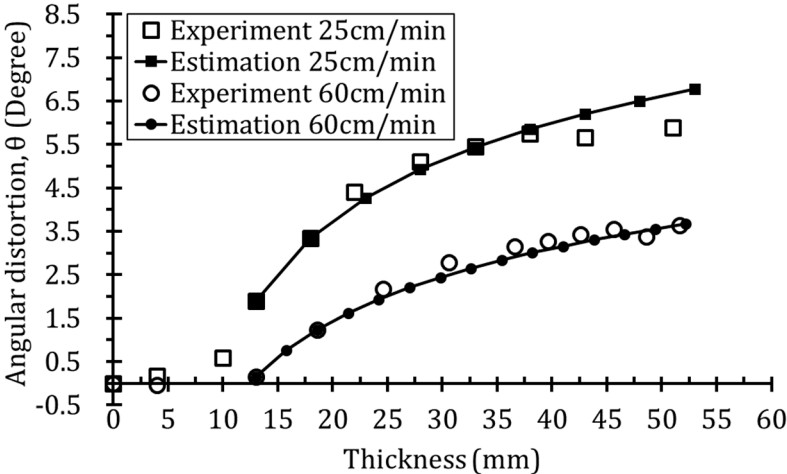

**Figure 12.** Comparison of experimental and predicted values of V butt welding experiment with 50 mm thickness for welding conditions of 280 A, 30 V, and varying travel speed.

The two prediction curves in Figure 12 were obtained by using the initial angular distortion, thickness, and degree of constraint. Although the curves suitably agree with the experiment results, the measured values showed smaller distortion than the predicted values over a bead that formed a thickness of approximately 40 mm for a travel speed of 25 cm/min. The prediction curves were calculated from experimental data before and after welding only one layer. Consequently, even small measurement fluctuations may cause a large error for estimating the degree of constraint. All the experimental data of angular distortion in Figures 10–12 show fluctuations, possibly by measurement errors or differences in the amount of weld deposition due to arc changes during welding. Thus, the prediction accuracy can be improved by acquiring more measurement points.

The X groove, or double-V groove, has a symmetrical shape with two V grooves about the weld root. When one-sided welding at the bottom is completed, as shown in Figure 13, the angular distortion on the upper side can be calculated while using Equation (11) by defining the filled thickness as initial thickness $h_0$ and setting it equal to root face $f$. Initial angular distortion $\theta_0$ is adopted as final value after filling the bottom (lower) side. The negative angular distortion is given when welding the opposite (upper) side.

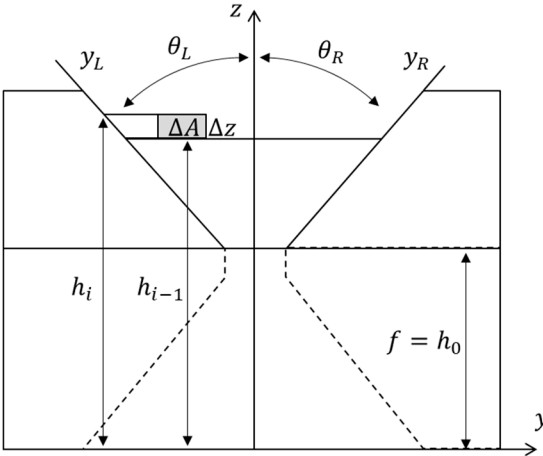

**Figure 13.** Geometry of X-groove butt welding joint.

Figure 14 shows a diagram of the specimen for the X butt welding experiments. We considered a welding current of 285 A and voltage of 29 V at varying travel speeds. The root gap was 4 mm and the root face was 0 mm. For the welding sequence, the upper side in Figure 14 was welded and filled first, and the lower side was welded after one pass was removed by arc gouging. The other welding conditions were the same as those for the previous experiments. The angular distortion was measured for each layer.

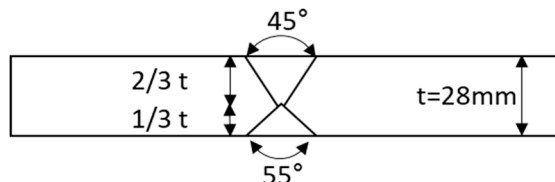

**Figure 14.** Diagram of specimen for X butt welding experiments.

The prediction model assumed that welding was resumed after removing one pass one layer by gouging when the bead thickness exceeded 18 mm, as it is not possible to predict the amount of gouging. Figure 15 shows the experiment and prediction results. When the bead thickness increased to 18.8 and 18.2 mm for travel speeds of 30 and 60 cm/min, respectively, it was reduced once to 14.9 and 15.4 mm due to gouging, respectively, corresponding to one layer height. The initial thickness and its angular distortion measured were 11 mm and 0.1° for welding speed of 30 cm/min, and 7 mm and 0.15° for 60 cm/min, respectively. The degree of constraint was calculated by inputting the angles that were measured before and after the third layer, being 1.04 for 30 cm/min and 1.45 for 60 cm/min. The degree of constraint increased with decreasing heat input, as with the prediction results for the V groove without strongback. When compared to the conventional V groove, there was a large prediction error of the final angular distortion after gouging, but the slope was similar for the X groove. This might be due to the amount of gouging, which is difficult to predict. Still, the overall trends suitably agreed, indicating that accurate prediction can be expected while using the proposed method for thicker plate welding under small change in distortion by gouging.

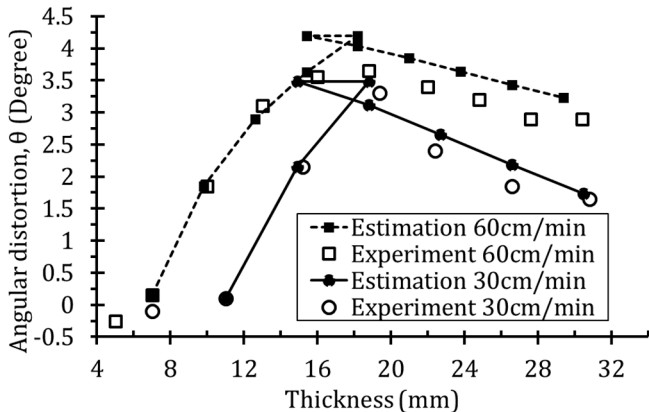

**Figure 15.** Comparison of experimental and predicted values of 28*t* X butt welding.

*3.3. Effect of Bead Size and Initial Thickness on Angular Distortion*

We investigated the multilayer welding characteristics while using the proposed model. Prediction curves of angular distortion with varying initial thickness and bead thickness were obtained. The target thickness was 50 mm, whereas the root gap and root face were set to 4 and 0 mm, respectively. The groove angle was 50° and the initial angular distortion was set to zero in all cases. Figure 16 shows the corresponding results, where the initial thickness corresponds to the origin of each curve. The bead height is related to the bead size, and it is given by each interval between consecutive marks on the curves. When the initial thickness changes from 4 to 10 mm, the angular distortion was substantially reduced to 61% (bead height: 5 mm) and 45% (bed height: 2 mm), respectively. When the initial thickness was the same, the angular distortion was smaller as the bead height (bead size) decreased, and the angular distortion was notably reduced to 91% (initial height: 10 mm) and 80% (initial height: 4 mm).

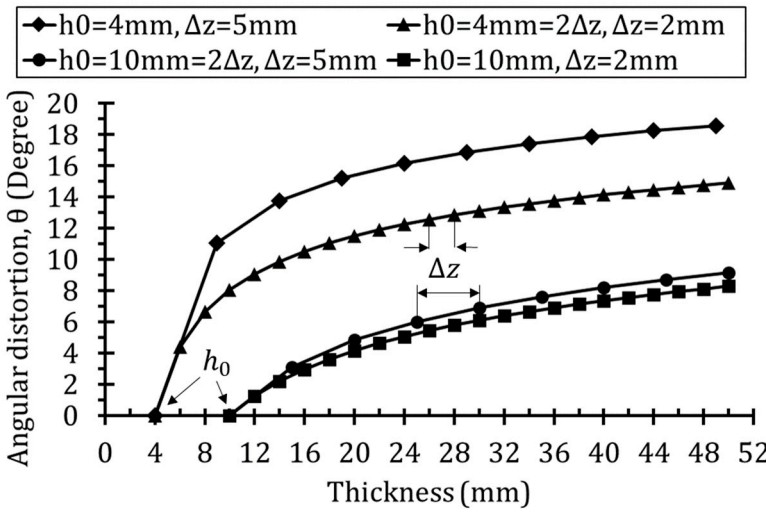

**Figure 16.** Prediction of angular distortion at different bead heights and initial thicknesses.

Higher heat input increases the bead size (height) and reduces the final angular distortion due to the increase in initial thickness for a fixed initial angular distortion under the same number of layers. On the other hand, if the initial thickness is the same, reducing the heat input to reduce the bead height should reduce the angular distortion. According to our previous study [10], the angular distortion decreases as the heat input increases during multilayer welding. In this case, the difference in initial thickness has occurred, because the same welding conditions were applied to all passes. In other words, when the initial thickness was twice the bead height (triangular and circular marks in Figure 16),

the final angular distortion decreases by the influence of the initial thickness. Overall, increasing the constraints reduces angular distortion. Alternatively, applying high heat input at welding onset for at least two layers for high initial thickness and then decreasing the heat input for small bead height can also reduce angular distortion during multilayer welding. This corresponds to the case where the initial angular distortion is zero. The initial angular distortion is difficult to predict, but it is desirable to minimize deformation during fitting, setting, and welding to the second layer through increased constraints.

## 4. Conclusions

Most of the structures for assembly are welded under unknown constraints. In this study, the angular distortion mechanism of welding joints was analyzed, and an analytical formula for its prediction with regards to the degree of constraint was developed. Experiments under varying heat input, thickness, and groove shape demonstrated suitable prediction accuracy. The main conclusions of this study can be summarized, as follows:

1.  The bead area was introduced instead of the heat input, and its relationship with angular distortion was determined and evaluated. The angular distortion is linearly proportional to the ratio of the bead area to the square of thickness, and it decreases after a threshold.
2.  An analytical formula for efficiently and accurately predicting angular distortion during multilayer butt welding was derived based on the above linearity and geometrical principle of the welding joint.
3.  For the prediction curve, the initial thickness and corresponding initial angular distortion should be obtained by the first measurement, and the degree of constraint by the second measurement.
4.  According to the prediction formula, the angular distortion decreases as the degree of constraint and root face increases and as the root gap and groove angle decrease.
5.  The degree of constraint significantly increased with strongback when compared to the free (unconstrained) condition. The difference in degree of constraint due to that in thickness of strongback was relatively small.
6.  Although no fixture is installed, the degree of constraint increases for small bead size. Fixtures can be used to increase the degree of constraint, increase the heat input up to at least two layers at welding onset, and subsequently keep a low heat input to minimize angular distortion in multilayer welding.

**Author Contributions:** Conceptualization, W.-J.S.; methodology, W.-J.S., S.-C.P. and H.-K.L.; investigation, W.-J.S., S.-C.P. and H.-K.L.; Validation, W.-J.S. and S.-C.P.; Formal analysis, W.-J.S.; software, W.-J.S.; data curation, W.-J.S.; writing—original draft preparation, W.-J.S.; writing—review and editing, W.-J.S.; visualization, W.-J.S.; Project administration, H.-K.L.; All authors have read and agreed to the published version of the manuscript.

**Funding:** This research received no external funding.

**Acknowledgments:** The authors acknowledge K. H. Yun in DSME Welding R&D Engineering Department for active support of the research.

**Conflicts of Interest:** The authors declare no conflict of interest.

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
