# Peer review of "Analytical Model for Angular Distortion in Multilayer Welding under Constraints"

_applsci, doi:10.3390/app10051848_

Round 1

Reviewer 1 Report

line 33

To complete the state of the art, I suggest inserting this type of work DOI: 10.1080/02726351.2017.1409850 Experiments and simulation of torque in Anton Paar powder cell.

line 52

To complete the state of the art, I suggest inserting this type of work DOI: 10.1115/1.4003099 Finite Element Analysis of Welding Processes by Way of Hypoelasticity-Based Formulation

line 99

To complete the state of the art, I suggest inserting this type of work: DOI: 10.1007/s11665-014-0905-z  Optimization and Prediction of Angular Distortion and Weldment Characteristics of TIG Square Butt Joints

line 184

Explain better how these thicknesses were chosen.

line 236

Explain better why from 0.35 onwards the decrease is linear.

line 317

How were the spaces between the three different septa determined?

line 333

It would have been useful to have a distortion angle of 1.5 to better validate the estimate.

line 337

Improve the representation of the initial part of the graph.

line 368

Explain better why the estimation trend is not respected.

line 411

It is suggested to change the type of representation the intersection between the lines does not facilitate understanding.

Author Response

Thank you for reviewing the manuscript. Answers to reviewer’s questions and the revised statements are included in the cell box as follows.

line 33

To complete the state of the art, I suggest inserting this type of work DOI: 10.1080/02726351.2017.1409850 Experiments and simulation of torque in Anton Paar powder cell.

Thank you for your suggestion. However, authors have decided not to include the reference you presented in the manuscript because it is not directly relevant to this study. Please respect authors’ decision.

line 52

To complete the state of the art, I suggest inserting this type of work DOI: 10.1115/1.4003099 Finite Element Analysis of Welding Processes by Way of Hypoelasticity-Based Formulation

The proposed paper derived the constitutive equation to consider the phase transformation and the transformation during welding, which this study does not cover. So, authors have decided not to include the reference you suggested in the manuscript. Please respect authors’ decision.

line 99

To complete the state of the art, I suggest inserting this type of work: DOI: 10.1007/s11665-014-0905-z  Optimization and Prediction of Angular Distortion and Weldment Characteristics of TIG Square Butt Joints

Although the proposed paper is related to angular distortion, we do not find any results that could be helpful for the background and method of this study. Since the logical flow does not fit, authors decided not to cite in this manuscript. Please respect authors’ decision.

line 184

Explain better how these thicknesses were chosen.

Thank you for your comment.

The following statement was included in the text.

The thicknesses of the bead-on-plate welding specimen were determined in consideration of the bead-formed thickness generated during the multilayer welding.

line 236

Explain better why from 0.35 onwards the decrease is linear.

Equation (9) derived from this study establishes a linear relationship, and Figure 6 shows the experimental results to prove this. There is a deviation, but the linearity is maintained below the critical point. Many researchers, as well as references [2] [3], claimed linearity within this range. In this manuscript, there is a sufficient explanation for this, so no contents are added.

line 317

How were the spaces between the three different septa determined?

The following statement was added in the text.

One strong back was installed in the center of the specimen, and two at each end where transient heat transfer and deformation occurs.

line 333

It would have been useful to have a distortion angle of 1.5 to better validate the estimate.

The angular distortion of 1.5 degree mentioned are not measurable because the value is between layers.

line 337

Improve the representation of the initial part of the graph.

Initial points in Figure 10 were corrected. For this, in the case of no strongback, the initial thickness for prediction was changed from 9.0mm to 8.7 mm as measured. After recalculation, the degree of restraint K changed from 1.04 to 1.09. Related values have been corrected in the manuscript.

line 368

Explain better why the estimation trend is not respected.

Some statement was revised as follows.

The prediction curves were calculated from experimental data before and after welding only one layer. Consequently, even small measurement fluctuations may cause a large error for estimating the degree of constraint. All experimental data of angular distortion in Figures 10 to 12 show fluctuations, possibly by measurement errors or differences in the amount of weld deposition due to arc changes during welding. Thus, the prediction accuracy can be improved by acquiring more measurement points.

line 411

It is suggested to change the type of representation the intersection between the lines does not facilitate understanding.

The upper line was changed to dotted line in Figure 15.

Reviewer 2 Report

The manuscript presented an analytical model to predict angular distortion in multilayer welding. This research topic dealing with the problems occurring oft in the actual shipyard is very important, and the results presented are enough. I recommend publishing the paper after answering the following questions.

  1. The abstract is not a minipaper. Please avoid including background information such as the sentences, “Multi-pass and multilayer welding is applied to construct … , but considerable effort and time are required due to the required thermo-elastic-plastic analysis for all welding passes.” Please start describing your work, “We propose an analytical model for fast prediction of angular distortion ….” It seems that “any welding joint” is not proper. Please reword it.
  2. Only angular distortion was considered in the study. However, there is a lack of explanation why it was so. Please add the information in the introduction, probably better at the very beginning.
  3. In equation (1), force and moment equilibrium equations are shown together. Please give a number per equation.
  4. Figures 3 -6 with their explanation and discussion could be better in the results and discussion section.

Author Response

Thanks for your keen review and comments. Answers to reviewer’s questions and the revised statements are included in the cell box as follows.

The manuscript presented an analytical model to predict angular distortion in multilayer welding. This research topic dealing with the problems occurring oft in the actual shipyard is very important, and the results presented are enough. I recommend publishing the paper after answering the following questions.

  1. The abstract is not a minipaper. Please avoid including background information such as the sentences, “Multi-pass and multilayer welding is applied to construct … , but considerable effort and time are required due to the required thermo-elastic-plastic analysis for all welding passes.” Please start describing your work, “We propose an analytical model for fast prediction of angular distortion ….” It seems that “any welding joint” is not proper. Please reword it.

Authors have revised the abstract as follows.

We propose an analytical model for fast prediction of angular distortion caused by practical multilayer (or multi-pass) butt welding under constraints. To this end, the relationships between angular distortion, bead size, thickness, and degree of constraint are derived by analyzing the welding deformation mechanism and considering bead-on-plate welding experimental results. Prediction curves are then obtained considering the geometry of the butt welding joint. We verify the formulas through experiments under various constraint conditions with different welding joint geometries, heat inputs, and thicknesses. The proposed model can not only predict angular distortion in butt joints of various shapes, but also allows to provide restraint methods and welding sequences for minimizing distortion.

2.Only angular distortion was considered in the study. However, there is a lack of explanation why it was so. Please add the information in the introduction, probably better at the very beginning.

The explanation was added as follows at the first paragraph.

For structures supporting compressive loads, the presence of initial out-of-plane deformation can cause buckling and lead to collapse in the worst case. Deformation in the welding of thick structures is largely due to transverse angular distortion, but controllable. Therefore, it is necessary to predict and minimize the distortion before construction.

3.In equation (1), force and moment equilibrium equations are shown together. Please give a number per equation.

They were separated and the corresponding equation numbers were revised.

4.Figures 3 -6 with their explanation and discussion could be better in the results and discussion section.

The explanation and discussion about Figure 3 to 6 could be included in the results and discussion section. However, authors determined to leave them as they are in the results section because they are also experimental results. Please respect the authors’ decision.

In addition, please note that the author’s previous work related to this study has been added as a reference as follows.

In line 70,

Seong et al. [11] introduced the concept of offset and constraint to develop a geometric based algorithm. The angular distortion of 145mm thick X-groove joint welding was predicted and compared with the finite element analysis results. Through the assessment procedure they proposed, the optimum welding sequence was provided.

Reviewer 3 Report

Subject

Authors developed a simple analytical formulation to predict the angular distortion of butt welded joints, specifying the limit of applicability and the procedure to be used for the evaluation of empirical parameters. The model has been applied to different joint configuration and the numerical prediction have been critically compared to experimental results.

The proposed model is interesting and overcomes the high-time consuming FEM models that constitute the traditional tool used for this purpose.

General observations

The subject is interesting and is suitable for a research article. Moreover, the methodology and the presentation of results are generally good.

For this reason, my suggestion is to accept the article. In order to improve the quality of the work, I suggest to the authors to specify better the following point.

The model is considered inapplicable when the ratio A/t2 is higher than the threshold value of about 0.375. If I understand correctly, the model is applied to multipass welding joints reported in section 3 only starting from the initial layer, which is identified by the height h0. In case of X groove, h0 is clearly defined by the geometry of the groove, whilst h0 is assumed in case of V groove. Could authors precise if the choice of h0 has been made to obtain a good agreement with experimental data or following other considerations? If it is, please add these considerations in the text.

Author Response

Thanks for your review. You pointed out the key points. Answers to reviewer’s questions and the revised statements are included in the cell box as follows.

Authors developed a simple analytical formulation to predict the angular distortion of butt welded joints, specifying the limit of applicability and the procedure to be used for the evaluation of empirical parameters. The model has been applied to different joint configuration and the numerical prediction have been critically compared to experimental results.

The proposed model is interesting and overcomes the high-time consuming FEM models that constitute the traditional tool used for this purpose.

General observations

The subject is interesting and is suitable for a research article. Moreover, the methodology and the presentation of results are generally good.

For this reason, my suggestion is to accept the article. In order to improve the quality of the work, I suggest to the authors to specify better the following point.

The model is considered inapplicable when the ratio A/t2 is higher than the threshold value of about 0.375. If I understand correctly, the model is applied to multipass welding joints reported in section 3 only starting from the initial layer, which is identified by the height h0. In case of X groove, h0 is clearly defined by the geometry of the groove, whilst h0 is assumed in case of V groove. Could authors precise if the choice of h0 has been made to obtain a good agreement with experimental data or following other considerations? If it is, please add these considerations in the text.

Thank you for the comment. It’s a sharp point.

Your understanding is correct. However, initial thickness h0 was not chosen to match the predicted and experimental values. Because of nonlinearity in angular distortion above the threshold, making deformation prediction practically difficult. Therefore, initial thickness must be set to the thickness of the second or more layers. All initial thicknesses were aquired by measurement, not geometrically determined. Both V and X groove cases shown in Section 3 were set h0 as the second or more layer thickness. The initial thickness of the V-groove in Section 3.1 has already been summarized ​​in Table2, and the initial thickness of the V-groove is also specified in the text in Section 3.2. However, in the case of 60 cm/min, the initial deformation amount was small and measurement was started from the third layer due to difficulty in measurement. Therefore, the following sentence has been added.

In line 331,

They were determined by measurement.

In line 363,

For welding speed of 60 cm/min, the initial thickness and angular distortion were 13 mm and 0.15° after third layer, respectively, and the measured distortion at the next layer was 1.23°. In this case, the change of the angular distortion was so small at the beginning that the measurement was started from the third layer.

In addition, the manuscript has already specified the limitation that the predicted value may vary depending on the initial thickness and angular distortion because of fluctuation of the initial angular distortion. If possible, the more the measured points after initial thickness, the more accurate the predicted value will be.